# A Novel Approach to Treat a Rare Case of Interprosthetic Humeral Fracture with Osteosynthesis and Combined Grafting: A Case Report and Review of the Literature

**DOI:** 10.3390/jfmk7040094

**Published:** 2022-10-26

**Authors:** Fabrizio Marzano, Valerio Pace, Marco Donantoni, Rosario Petruccelli, Paolo Ceccarini, Auro Caraffa, Lorenzo Maria Di Giacomo

**Affiliations:** 1Trauma & Orthopaedics Department, Santa Maria della Misericordia Hospital, University of Perugia, 06024 Perugia, Italy; 2Azienda Ospedaliera Santa Maria Terni, 05100 Terni, Italy

**Keywords:** bone fragility, humeral fracture, periprosthetic humeral fracture, interprosthetic fracture, bone grafting

## Abstract

Interprosthetic humeral fractures (IHFs) are severe injury patterns associated with surgical issues and contradictory results. The knowledge and literature on this topic are still lacking. A 76 year-old woman was treated for a fracture occurred between the shoulder and elbow stemmed prosthesis. Severe bone loss was associated with the fracture. Treatment: Open reduction, plate fixation, and bone grafting were considered. A xenograft (used as a mechanical strut medially), a synthetic graft associated with bone growth factors, and scaffolds improved the bone healing process. Satisfactory clinical and radiological outcomes were obtained. A scoping review of the literature was also performed by the authors. Only eight papers reported IHFs with a low level of evidence. In total, eight patients were treated; one paper that reported on biomechanical aspects using finite element analysis is discussed. Conservative treatment leads to non-union, and the surgical approach is the gold standard. The osteosynthesis technique associated with bone grafting leads to the best outcomes. The use of a xenograft mechanical strut, associated with synthetic biological bone grafting, led to complete bone union at 9 months follow-up. Larger cohorts, more standardised results, and multicentric studies are mandatory in order to improve and establish a management and treatment algorithm.

## 1. Introduction

Interprosthetic humeral fractures (IHFs) are a rare condition, considered difficult to manage, and require challenging techniques and strategies for appropriate treatment and results. The increasing trend in shoulder arthroplasty implants in the last decade [1,2] and the development of better elbow implants and surgical techniques [3,4], associated with a worldwide ageing process, will lead to more frequent cases of this complicated injury pattern in the future. The issues of IHFs are thought to be related to the following aspects: the pattern type of the fracture, bone quality, the proximity and stability of the prosthesis, and stress risers due to endomedullary components [1,2,3,4]. While for femoral fractures between the total knee arthroplasties (TKA) and the total hip arthroplasties (THA), knowledge is developing [5,6], to date, the literature describing IHFs and their treatment and outcomes is really poor [1,2]. We describe a very rare case of a complex fracture between a stemmed cemented reverse shoulder arthroplasty and elbow arthroplasty associated with severe bone loss. The novel treatment consisted of open reduction and osteosynthesis with plating and cerclage associated with xenografting from the equinus femur associated with biological bone grafting with adjunct growth factors. This led to subjectively and objectively good functional and radiological outcomes. Additionally, a scoping review of the literature in relation to IHFs was performed. The aim of our paper is to report our experience with this rare condition and the good results achieved by our unique surgical strategy to use equinus xenograft plus bone graft enriched with growth factors. We also aim to review the current literature on the studied topic in order to highlight the current trends and stimulate further research in the field.

## 2. Case Report

A 76-year-old woman was transferred from a primary general referral centre to our secondary Trauma and Orthopaedic Department referral centre after sustaining a low-energy fall. She fell onto the right side when at home. The patient’s medical history included several comorbidities: type II diabetes mellitus, severe osteoporosis, hypertension, hypercholesterolemia and hyperuricemia, complex treated trauma to her right shoulder and elbow (2010 and 2012), respectively, with reverse shoulder arthroplasty and total elbow arthroplasty (at a different orthopaedic trauma centres).On admission, plain radiographs of the right arm showed an IHF between the two stemmed prostheses (Figure 1).

The fracture occurred at the shoulder’s stem tip without compromising the stability of the arthroplasty associated with cement leakage. The fracture was classified as type D according to the Unified Classification System (UCS) [7]. Type D is a fracture affecting one bone which supports two replacements, such as the humerus following shoulder and elbow replacement, or the tibia following knee and ankle replacement. [7] The shoulder prosthesis was a reverse implant with a cemented stem, whilst the elbow arthroplasty was an uncemented Conrad Moore implant. Moreover, a severe poor bone stock was noted around the fracture site. Clinical examination revealed a high level of right arm pain and tenderness; a vast ecchymosis without any skin lesion and no signs of vascular or neurological injury were noted. The vital signs and the laboratory results were in the normal range despite the several comorbidities. The literature related to the management of such injuries was reviewed within the local Trauma and Orthopaedic Department, and the case was discussed at the local multidisciplinary meeting: The decision to proceed with surgical management was agreed upon (in agreement with the patient), and the treatment strategy consisted of open reduction, internal fixation with plating and cerclage, and xenografting from the equinus femur and biological bone grafting with adjunct growth factors in case of poor bone quality and stock. One week after the admission (after having held the aforementioned meetings and after the arrival of the surgical instrumentations and the graft), the patient underwent the planned surgical procedure (Figure 2).

The patient was set up in a beach chair position. An extended deltopectoral approach was chosen; a careful, blunt dissection of the surrounding soft tissues was then performed, proceeding deeply into the muscular part of the biceps brachii and brachialis, both of which showed a high grade of contusion. The radial nerve was individualised at the spiral groove and laterally protected throughout the surgery. The ulnar nerve was also visualised and isolated and protected throughout the procedure. At the level of the fracture, the cement leak was noted and removed, and the tip of the shoulder stem was tested and appeared to be stable. Due to very poor bone quality, a synthetic bone graft associated with growth fracture in a mesh form and a strip form (Bone Graft I-Factor, Cerapedics) was inserted at the fracture site, and the supportive medial bone graft from the equinus femur was also inserted as a mechanical strut (Bioplant-Activia, 150 × 20 × 6 mm equine bone graft). The fracture was reduced and stabilised under fluoroscopy with a 4.5/5 mm eight-hole reconstruction plate, further stabilised by four cerclages (DePuy-Synthes cerclage system) with a tension of 45 Nm. The stability was tested, and the fixation was deemed to be stable. Accurate haemostasis was provided throughout the surgical procedure. Wound closure was performed in layers. A Gilchrist bandage was applied. One blood transfusion was administrated on the second postoperative day, and no other perioperative complications occurred during the hospitalisation. A routine wound check was carried out till clip removals. The post-op laboratory results were in the range after the second post-op day. The postoperative X-rays on the first postoperative day showed good alignment and stability of the construct (Figure 3).

The rehabilitation program included 30 days of shoulder immobilisation whilst keeping a Gilchrist bandage. Only the active mobilisation of the elbow was encouraged from the first postoperative day. Magnetic field therapy (Biostim, IGEA Clinical Biophysics) was also prescribed in order to improve and speed up the bone healing process. A full range of motion (without restrictions) was allowed after 30 days of radiological and clinical follow-up, but muscle strengthening against resistance was forbidden at the time and was gently allowed at 3 months after surgery. A periodical clinical and radiological follow-up was carried out each month in the first trimester and then every three months up to a year. At 6 months of follow-up, the wound was well-healed, and the range of motion was gradually improving. The check X-ray showed a good level of bone healing with the stability of metalwork and implants. The radiological healing was achieved at 9 months post-op. The X-rays at 9 months post-op showed good alignment and complete integration of the graft strut. The clinical and functional results were satisfactory both for the elbow and the shoulder (Figure 4). At the last follow-up, the DASH score was 75, the NRS (numerical rate score) was 2, the shoulder arc of motion was 100° in elevation, the external rotation was 5°, and the internal rotation was toward the gluteus, and the elbow arc of motion was in ex/flex 30–110 and 50–45 in prono-supination. At 1 year follow-up, the check X-ray was again satisfactory, with no reported issues. The clinical findings were similar to those reported at 9 months of follow-up. The patient is satisfied with the postoperative course, and her expectations were met.

## 3. Literature Review Strategy

A literature review was performed based on four relevant databases (Cochrane, Scopus, PubMed, and Google Scholar). The terms of research were “interprosthetic” and “peri-implant” and “humeral fracture” and “periprosthetic elbow arthroplasty” and “periprosthetic shoulder arthroplasty”. The research was performed by two of the authors. No languages or time restrictions were applied. The exclusion criteria were peri-implant humeral fracture (the fracture that occurred in the humerus with at least one osteosynthetic device) and infections. Due to the paucity of data on this topic and the poor consistency of the reported cases and evidence, a systematic review could not be carried out. Therefore, a narrative review was carried out instead.

## 4. Results

Only eight papers discussing the management and treatment of IHFs [8,9,10,11,12,13,14,15] were found through the literature review and were included in our article. A biomechanical study with a finite element analysis was also included in our review [16]. The studies and their characteristics are listed and described in Table 1. The year range of the included articles varied from 1999 to 2019. All the studies were case reports or technical note articles, with relatively low levels of evidence. Only one biomechanical study based on finite element analysis was found. A total of eight patients were treated, (six women, one man, and one case of unknown gender). The age range was from 43 to 71 years. Three cases were reported in patients affected by severe rheumatoid arthritis, and one case of severe haemophilia was also indicated. In the rest of the papers, comorbidities were not reported.

The fractures occurred between two stemmed prostheses; in one case, the fracture proximally extended up to the shoulder, and in two cases, to the tip of the elbow arthroplasty. In 1 case an insufficiency fracture was suspected due to fracture pattern and characteristics. No papers reported any classification system. The preferred surgical approach was posterior with the isolation and protection of radial and ulnar nerves throughout the procedure. One case described a conservative strategy through bracing, whilst in one case, the management was not appropriately described. The most utilised fixation technique involved a plating system, used in five cases and associated or not with wiring and cable. Four cases necessitated bone grafting, which was allograft in all the cases. Two cases reported the use of the APC (allograft prosthetic composite) technique with tibia allograft.

The consolidation times for fractures were reported in four papers, ranging from 7 months to 1 year. No consolidation occurred in one case. Complications were reported only in one paper in a patient with postoperative radial palsy. The functional score was reported only in one article, according to which the DASH score was 54.1, and the self-evaluation score was 10 of 10. A return to activity daily living was reported in two articles with a return to ADL (activity daily living) in 10 months and 6 weeks after surgery, respectively.

## 5. Discussion

IHF is a rare and challenging condition. Its management requires the best available surgical equipment in addition to excellent surgical skills and experience. We reported a rare IHF case with unique features and surgical management, treated and followed up at our department. The main encountered difficulties included the fracture pattern, the management of severe bone loss, the choice of the surgical approach and technique, and the fixation equipment. We successfully treated the IHF patient with a novel approach and technique including the retention of the two stable implants, a plating system, a biologic synthetic graft, and a xenograft with growth factors. Our strategy allowed for very good osteointegration and fracture healing at 9 months of clinical and radiological follow-up. Our literature review highlighted significant inconsistencies with regard to the fracture pattern, surgical management, results, rehabilitation program, and follow-up. Moreover, we would like to stress that a lack of standardisation in relation to management and treatment was also noted. Indeed, our research found only eight papers in the included databases, the methodology and results of which showed a worrying level of heterogeneity that could be linked to a very low level of evidence.

The first description of an IHF case goes back to Gill et al. in 1999 [15], observing the outcomes of 17 patients affected by rheumatoid arthritis treated with arthroplasties. In this case series, the authors observed the presence of two cases of fractures that occurred between two prostheses. Unfortunately, the treatment strategy for such cases was not presented, nor was it discussed in the article.

The periprosthetic fracture classification systems for the shoulder and elbow are well-known [17,18,19,20,21,22,23], but previous studies did not consider the “inter-implant” fracture. Therefore, we advocate more widespread use of the UCS system [7], as it seems to be the most reliable and relevant classification system as things stand. Alternatively, a specific classification system should be studied, developed, and introduced.

Among the major issues for IHF cases, we must consider the fracture pattern, implant stability, and bone loss and stock.

The most frequent fracture type is reported to be a short oblique or transversal fracture between the two implants, mostly occurring after low-energy falls and in cases of severe osteopenia [8,9,12,15,20]. Therefore, an important role seems to be related to the stemmed components and their stress risers.

Gill et al. [15] recommended some strategies to be used in order to reduce the stress risers over the unfilled humeral segment, which acts as a point of weakness. If standard-length shoulder and elbow components are used together in the same humerus, bone cement should bridge the small distance between the prostheses. If shorter humeral components are used, a cement-restriction device should be used to ensure a long length (approximately sixty millimetres or more) of the unfilled humerus between the cement columns. Inglis et al. [12] recommended the use of long-stemmed elbow and humeral implants in order to minimise the bone stress risers, or alternatively, to fill the remaining humeral canal with a cement column. These considerations lacked experimental tests and scientific evidence. However, they are very thoughtful considerations coming from very experienced surgical study groups and should definitely be taken into account through the planning process when dealing with IHFs.

Plausinis et al. [16] carried out a study with a 3D finite element analysis in order to prove the correlation between the bone bridge and the effect of cement canal filling with the increased risk of IHFs. They concluded that the length of the bony bridge had no effect on the stresses on the humerus and that filling the humerus with cement does not decrease the stress on the humerus but could even be an issue in case of revision. In cases with intact cortical bone, the risk of a periprosthetic fracture is independent for each prosthesis [14,16].

An appropriate and tailored surgical management (based on the specific fracture pattern, bone stock, and the patient’s comorbidities) seems to be the gold standard of treatment for the best possible results. Inglis et al. [12] reported their attempt to conservatively treat IHFs, which resulted in non-union. All other authors preferred surgical management. A posterior approach was used in most of the cases, utilising the tricep splitting method [8,9,13,14] or the tricep sparing technique [11]. The most popular choice was the use of an LCP (locking compression plate) with allograft, aided by cerclage and cables.

Carroll et al. [8] described a technique with a 90–90 double plating in the distal humerus, used in order to enhance the stability of poor bone quality. LeBlanc et al. [11] described a novel approach with allograft prosthetic composite (APC) with tibia allograft without hardware augmentation in a severe haemophiliac patient, with good final results. The procedure led to allograft integration and a satisfactory range of motion, although the patient was found to have postoperative partially radial nerve palsy, presumably due to the bleeding caused by surgical traction in the haemophiliac patient.

Taking into account the related scientific evidence and the specific aspects related to our case, we decided to proceed with an extended deltopectoral approach in a beach chair position. By using this strategy, we were able to use the previous incision without any further soft tissue damage, allowing an extensile approach in order to individualise and protect the radial and ulnar nerves throughout the procedure; our approach also allowed us to fill the lost bone with bone graft and achieve a satisfactory reduction and synthesis.

Bone substitutes are essential and strictly necessary in these types of surgeries, due to the low bone quality. They allow for a quicker and more solid fracture healing process. Allograft substitutes are the most commonly used for the treatment of such injuries [9,10,11,14].

We used an I-Factor Bone Graft (Cerapedics) in order to enhance bone healing. The I-Factor is a composite bone substitute biologic material consisting of a P-15 synthetic collagen fragment adsorbed onto an inorganic bone mineral suspended in an inert, biocompatible hydrogel carrier. A P-15 Osteogenic Cell Binding Peptide is bound to an inorganic bone mineral (ABM). This unique combination creates a surface-bound “attract, attach, and activate” mechanism of action that enhances the body’s natural bone healing process. Being surface-bound, all cellular activity resulting from the P-15 Osteogenic Cell Binding Peptide attachment is restricted to the implant surface, so the bone cannot grow where it does not belong (ectopic bone growth). The outcomes in spinal surgery showed high safety and effectiveness and similar results to those of the cases where autografts were used [24,25,26]. Moreover, it provides more benefit/cost rather than the association of allograft and growth factor. The I-Factor Bone Graft is considered an advanced biologic supported by level I evidence and a very good cost/benefit ratio. Furthermore, it is not a morphogen and only activates those cells that are pre-programmed to become osteoblasts.

A bioplant is a synthetic bone graft. After initial osteointegration, the graft increases load resistance and bone density, due to its shape that provides scaffolds for bone ingrowth. These types of grafts are derived from the equinus bone through a selective enzyme denaturation process and work at 37°. This allows for the elimination of organic components and the retention of the mineral and collagen components. Furthermore, the selective denaturation system is able to retain the growth factors and their activity. Osteoclasts recognise the mineral components as endogens, and this triggers a bony remodelling process. The collagen component is able to provide adequate strength and elasticity to the bone and co-activate the endogenous growth factors, in addition to representing the natural substrate for osteoblast adhesion. All these activities act together contributing to forming a favourable and physiological environment for bone growth.

Although not biomechanically confirmed, in our opinion, the use of a long plate that proximally and distally extends by at least two-cortical diameters beyond the fracture is fundamentally necessary to avoid stress at the level of the fracture site, whose presence could increase the risk of osteosynthesis failure.

Appropriate fracture healing requires at least 6 months and up to 1 year [8,9,13,14]. We reported full fracture healing at 9 months of follow-up.

The reported outcomes are really different and contradictory among the articles included in our review. Mavrogenis reported a return to ADL (activity daily living) in 10 months [14], while Kieser et al. [9] reported a DASH score of 54.2 at 1 year, and LeBlanc and DeFroda reported only the elbow range of motion [11,13]. Our patient is now regularly followed up, with satisfactory radiological and clinical outcomes. The X-rays at 9 months post-op showed good alignment and complete integration of the graft strut. The clinical and functional results were satisfactory both for the elbow and the shoulder (Figure 4). At the last follow-up, the DASH score was 75, the NRS was 2, the shoulder arc of motion was 100° in elevation, the external rotation was 5°, the internal rotation was toward the gluteus, and the elbow arc of motion was in ex/flex 10–110 and 50–45 in prono-supination. These can objectively be considered very good final results, considering the previously reported cases.

The limitations of our study are strictly related to its nature. Due to the rarity of the condition, we are not able to lead studies with wider cohorts at the moment.

We would like to stress again that there is a paucity of data on the studied topic and poor consistency of the reported cases and evidence. We found mostly case reports with significant heterogeneity and obvious low levels of evidence. We advocate the need for studies with a higher level of evidence and with larger cohorts, in order to achieve international consensus on the treatment algorithm. A specific classification system is another necessity.

In this environment of a lack of scientific evidence, we were able to review the available literature and decide on a good surgical strategy (with multidisciplinary consensus) that allowed very good radiological and functional results. This is the first time that our orthopaedic team performed such a combination of surgical techniques. This is simply because such a complex diagnosis was never encountered by the team. However, we foresee more similar cases in the future due to epidemiological aspects (the ageing population and the increased number of patients with shoulder and elbow prosthetic surgery). Our proposed technique seems to be relevant and appropriate to treat similar cases in the future, and we strongly recommend our strategy plan to all those surgeons dealing with this pattern of injury. Our novel technique with the use of an equinus xenograft plus biologic bone grafting enriched with growth factors showed encouraging results and can be considered a very good management strategy tailored for the characteristics of our patient. However, more cases and longer follow-ups are necessary to establish and finally validate the success of the presented treatment. We already presented our case, technique, and results in our periodical regional orthopaedic meetings in order to make our strategy plan available for patients presenting with this pattern of injury. We hope for further dissemination of the presented results in order to make our novel technique available for all orthopaedic teams dealing with similar cohorts.

## 6. Conclusions

IHFs are rare events that need accurate multidisciplinary planning, preparation, and good surgical skills in order to achieve good outcomes. Osteosynthesis with appropriate bone support and healing promotion is mandatory to succeed. The aid of scaffolding and growth factors might induce a faster healing process and, therefore, better outcomes.

We presented a unique case of IHF successfully treated with open reduction and osteosynthesis with plating and cerclage associated with a xenograft from the equinus femur associated with biological bone grafting with adjunct growth factors. 

We advocate and encourage further research in order to achieve a standardised classification system and treatment algorithm.

## Figures and Tables

**Figure 1 jfmk-07-00094-f001:**
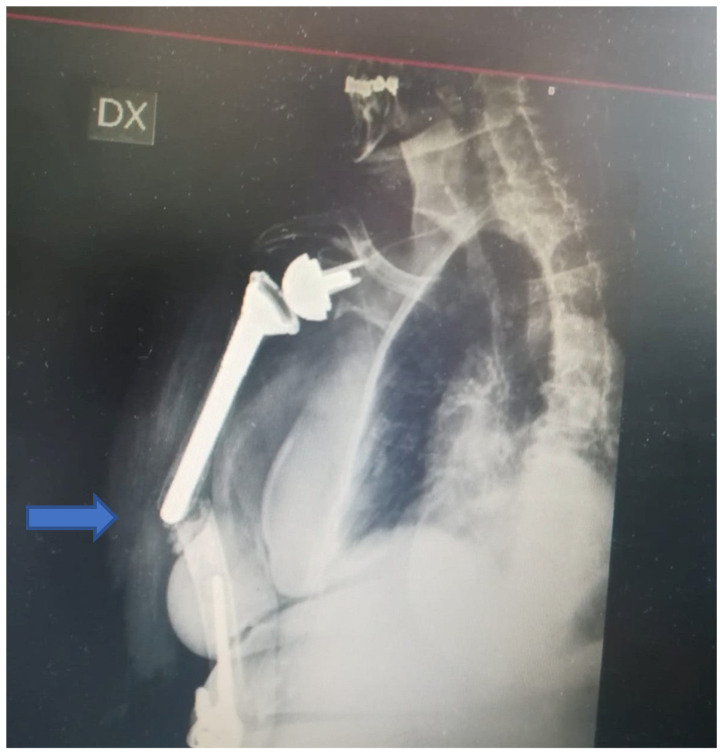
X-ray at admission. Complex fracture between a stemmed cemented reverse shoulder arthroplasty and elbow arthroplasty associated with severe bone loss.

**Figure 2 jfmk-07-00094-f002:**
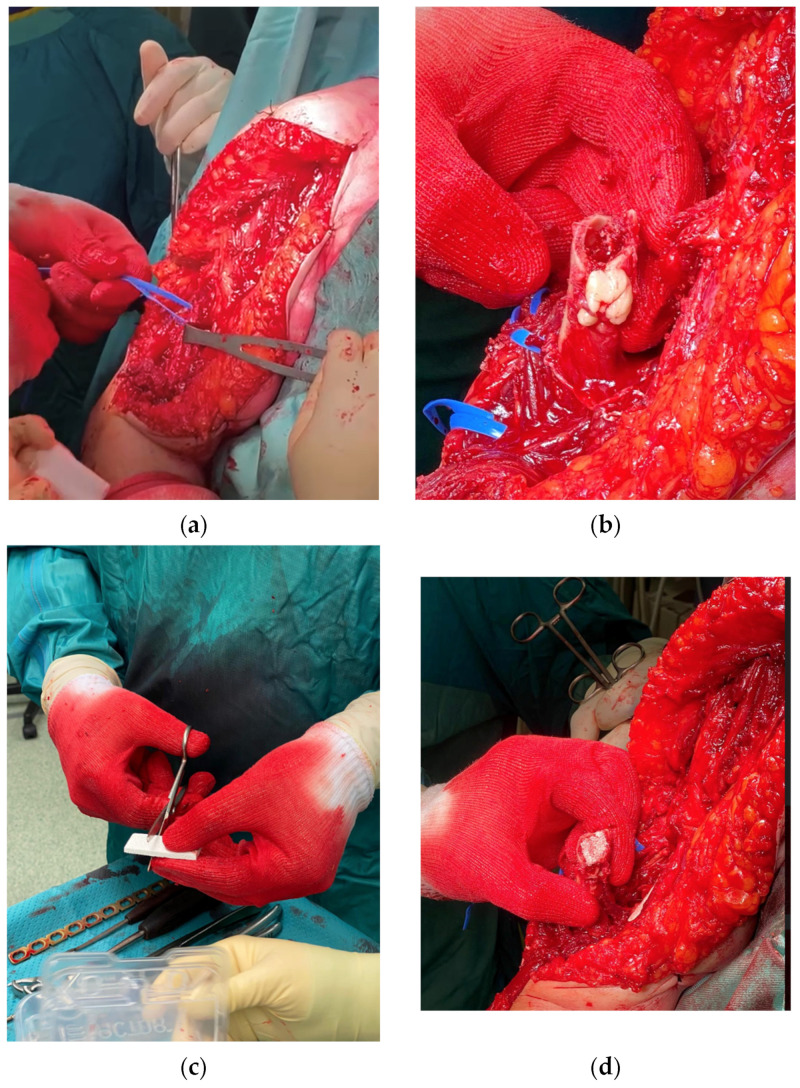
Treatment of IHF and its surgical steps: (**a**) extended deltopectoral approach and isolation of radial nerve; (**b**) removal of cement leakage from previous surgery; (**c**,**d**) preparation and insertion in the humeral canal of synthetic bone as a scaffold for bone ingrowth and growth factors; (**e**,**f**) Synthesis of the fracture, equinus graft medially as mechanical strut stabilised through plate and cerclages and intra-operative fluoroscopy after strut positioning.

**Figure 3 jfmk-07-00094-f003:**
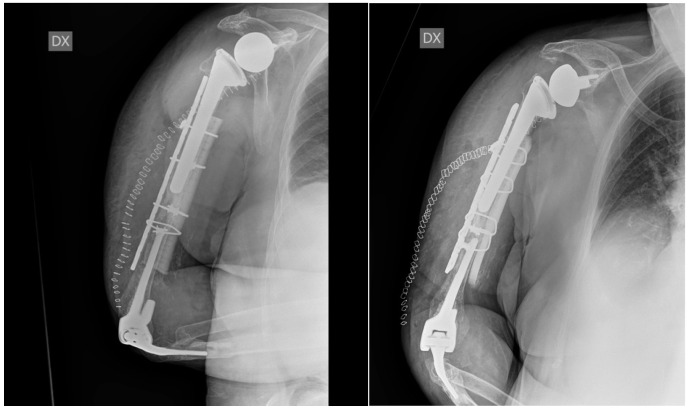
Postoperative X-ray at 1 day. The X-rays show good reduction and alignment of the fracture, good fixation and strut position, and stability of the previously implanted shoulder and elbow prostheses.

**Figure 4 jfmk-07-00094-f004:**
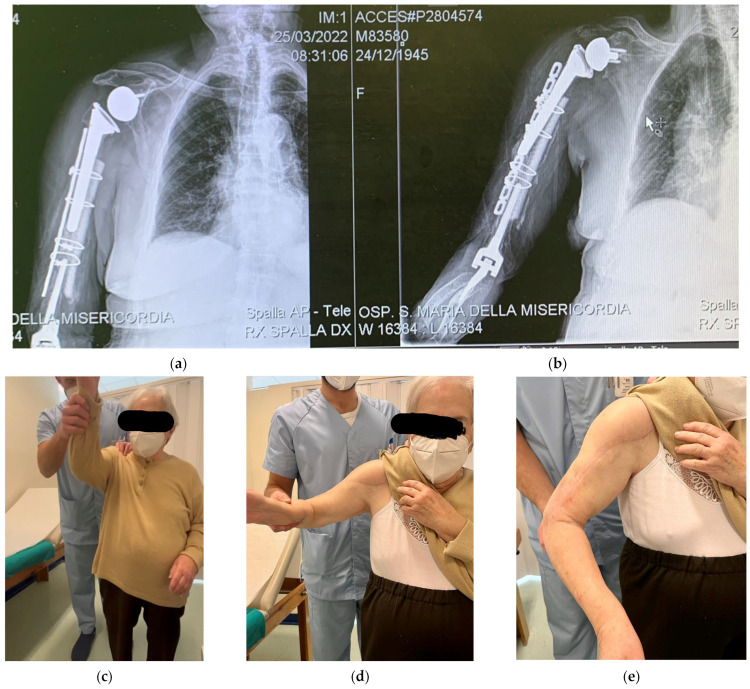
Radiological and clinical follow at 9 months post-op: (**a**,**b**) the X-rays show stability of the fixation metalwork and of the shoulder and elbow prosthesis, good bone healing process, and good strut integration; (**c**–**e**) satisfactory shoulder range of motion; no scare issues.

**Table 1 jfmk-07-00094-t001:** Papers included in the review regarding IHFs and their treatment.

Author (Year)	Type of Study	n° Patient	Treatment	Healing	Complications
Gill (1999) [15]	Case series	2	Not reported	Not reported	Not reported
Inglis (2000) [12]	Case report	1	Conservative	Not achieved	Not reported
Carroll (2008) [8]	Case report	1	90°–90° double plating	6 months	None
Kieser (2011) [9]	Case report	1	Allograft prosthetic composite and plating with cable	Clinical at 1 year	Not reported
Mavrogenis (2011) [14]	Case report	1	Posterior plating and allograft	10 months	None
LeBlanc (2012) [11]	Case report	1	Allograft prosthetic composite	Not reported	Postoperative radial nerve palsy
Grechenig (2017) [10]	Technical note	1	Plating, cable, and allograft	Not reported	Not reported
DeFroda (2019) [13]	Technical note	1	Double plating	1 year	Not reported

## Data Availability

Not applicable.

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
