# Peer review of "A Novel Approach to Treat a Rare Case of Interprosthetic Humeral Fracture with Osteosynthesis and Combined Grafting: A Case Report and Review of the Literature"

_jfmk, 2022, doi:10.3390/jfmk7040094_

Round 1

Reviewer 1 Report

It is valuable to readership since this condition is rare. It is a valuable case report.

Author Response

Thank you very much for your very valuable and expert review. All auhtors much appreciate your review. We are delighted you consider our paper as a valuable article.

Kind regards

Reviewer 2 Report

The Case Report entitled  “A novel approach to treat a rare case of Interprosthetic Humeral Fracture with Osteosynthesis and combined grafting: a case report and review of the literature” presented by Fabrizio Marzano  , Valerio Pace , Marco Donantoni , Rosario Petruccelli , Paolo Ceccarini , Auro Caraffa  and Lorenzo Maria Di Giacomo illustrates a case of Interprosthetic Humeral Fractures (IHF) in a  76- year old woman. They describe new surgical approach to treat a complex fracture between a stemmed cemented reverse shoulder arthroplasty and elbow arthroplasty associated with severe bone loss. The authors propose a novel surgical strategy consisting of open reduction and osteosynthesis with plating and cerclage associated with a xenografting from equinus femur associated with biological bone graft with growth factors enriched with growth factors. The authors describe good functional and radiological results derived from the treatment and discussed their results compared with those reported in the literature on IHF.

The case presentation is interesting. The presentation of clinical and surgical data is meticulous and translates an extensive experience of the authors. In my view, the strength of the study is the precision in the description of the surgical method. On the other hand, the weak point is the literature review that could be more complete. Thus, a more complete comparative analysis of the results obtained in the present study in relation with those in the scientific literature could be performed. As a minor point, the legend of the figure 4 does not contain sufficient information to be comprehensive; a brief description is needed to explain the photographs.

After these minor revisions, the manuscript could be accepted

Author Response

Thank you very much for your very expert review of our submitted paper. All authors appreciated your very valuable inputs and we have made changes accordingly. We could not widely expand the comparative analysis of the results obtained in relation with those in the scientific literature as we could not find further relevant articles in the literature on the topic. A part from that, all advices have been strictly followed. We have added relevant articles in the bibliography.

Reviewer 3 Report

Page 2, Figure 1: It would be beneficial to the reader to put an arrow on the image pointing to the IHF in the X-ray.

Page 2, line 58: A brief description of what type D is would benefit the reader who is not intimately familiar with the classification system.  Many readers will be those in bone research but may not be a clinician or an expert who knows the system.

Page 5, figure 4: It may be worth blacking out the eyes in the far left image as well.

Overall: what is the longest period of follow up in this publication?  It would be good to include follow up at 6 months and 1 year post-op or whatever time frame is standard in ortho surgeries.  I see 9 months referenced on page 6.  I would say hold publication and include a 1 year summary too.

Overall: Has the team performed any additional surgeries like this one or does the team have plans to perform another one in the near future?  Has the team partnered with nearby physicians so if a similar case appears, the community of doctors know this technique expertise is nearby?  

Author Response

Page 2, Figure 1: It would be beneficial to the reader to put an arrow on the image pointing to the IHF in the X-ray.

WE HAVE ADDED THE ARROW ON THE PICTURE.

Page 2, line 58: A brief description of what type D is would benefit the reader who is not intimately familiar with the classification system.  Many readers will be those in bone research but may not be a clinician or an expert who knows the system.

THANK YOU VERY MUCH, WE AGREE WITH YOUR POINT. WE HAVE ADDED A DESCRIPTION OF TYPE D FRACTURES.

Page 5, figure 4: It may be worth blacking out the eyes in the far left image as well.

APOLOGIES FOR THIS, WE UPLOADED AN OLD VERSION OF THE PICTURE. WE HAVE NOW UPLOADED THE CORRECT VERSION WITH BLACKED EYES.

Overall: what is the longest period of follow up in this publication?  It would be good to include follow up at 6 months and 1 year post-op or whatever time frame is standard in ortho surgeries.  I see 9 months referenced on page 6.  I would say hold publication and include a 1 year summary too.

THANK YOU VERY MUCH FOR THIS VALUABLE INPUT. WE HAVE NOW ADDED DETAILS OF OUR FOLLOW UP, UP TO 1 YEAR.

Overall: Has the team performed any additional surgeries like this one or does the team have plans to perform another one in the near future?  Has the team partnered with nearby physicians so if a similar case appears, the community of doctors know this technique expertise is nearby?  

THANK YOU VERY MUCH. WE HAVE CLARIFIED IN THE DISCUSSION SECTION ALL THE ABOVE POINTS.